# Establishing consensus on the implementation of Anticoagulation Stewardship Program with cardiologists in Pakistan: A Delphi study

Wajiha Razzaq, Muhammad Atif *, Kanza Arshad, Imran Masood

Department of Pharmacy Practice, Faculty of Pharmacy, the Islamia University of Bahawalpur, Bahawalpur, Pakistan

* pharmacist_atif@yahoo.com

## Abstract

### Background

The Anticoagulation Stewardship Program (ASP) improves clinical outcomes, promotes patient safety, and supports healthcare systems in delivering high-quality, evidence-based anticoagulation management. This study aimed to develop a consensus among cardiologists about the implementation of the ASP Program in Pakistan.

### Methods

A three-round Delphi study was conducted utilizing an online questionnaire. In Round 1, cardiologists (Delphi panel experts) reported their consensus with the items in the questionnaire using a 3-point Likert scale. The selection of items for Round 2 was based on acceptance by ≥66.6% of the cardiologists and the agreement of the scientific committee. In Round 2, the panelists assessed those items that failed to gain consensus in Round 1. In Round 3, a face-to-face meeting was conducted among the scientific committee to evaluate the items that failed to gain expert agreement in Round 2 to form the final consensus document. Descriptive statistics was used to present the data.

### Results

A total of 90 cardiologists from 30 hospitals were invited to participate in the study. Of these, 75 agreed to participate in Round 1 of the study (83% response rate). 68 cardiologists completed the survey in Round 2. Initially, 33 items from four domains were evaluated by the Delphi experts in Round 1. 18 items reached consensus in Round 1, 4 items reached consensus in Round 2, and 2 items reached consensus in Round 3. The final consensus document comprised 24 items. The study showed absolute consensus among national cardiologists regarding the implementation of ASP in Pakistan. A considerable agreement was achieved regarding essential

**Data availability statement:** All data relevant to findings of the study is available in the manuscript itself, and uploaded as supplementary information.

**Funding:** The author(s) received no specific funding for this work.

**Competing interests:** The authors have declared that no competing interests exist.

components needed to strengthen ASP for anticoagulation management in the cardiology departments.

## Conclusion

This study emphasized the need for educational sessions for patients and healthcare professionals, collaboration with healthcare authorities, and allocation of financial resources for ASP. This study also identified consensus among cardiologists on the perceived benefits of ASP for patients and the healthcare system. Several barriers that hindered the implementation of ASP in Pakistan were identified, including patient- and healthcare system-related barriers.

## Introduction

A large number of patients utilize anticoagulant therapy globally to manage cardiovascular diseases such as atrial fibrillation, coronary heart disease, congestive heart failure, heart attack, and venous thromboembolism [1]. A meta-analysis of studies on anticoagulant use reported that there was increase in the use of oral anticoagulants from 2010 to 2018 [2]. Contemporaneously, there has additionally been a substantial increase in adverse drug events (ADEs) related to anticoagulants, including high-risk bleeding and thromboembolic complications [1,3]. This necessitates the careful use of anticoagulants, considering the fact that prescribing practices and patient attributes such as age, co-morbidities, concomitant use of other medications, and pharmacogenetics highly influence the incidence of ADEs [4].

The notion of stewardship programs started in the late 20th century to promote judicious use of antimicrobials. The antimicrobial stewardship program remained very effective at reducing the occurrence of adverse drug reactions (ADRs), healthcare costs, antimicrobial resistance and improving clinical outcomes among patients [5]. In 2014, the ADEs associated with anticoagulants surpassed antibiotics as a leading cause of emergency department visits in the United States (US) [6]. The US National Action Plan for ADE prevention (2014) also documented that anticoagulants were not only the leading cause of emergency department visits among outpatients but also caused ADEs in 10% of inpatients on anticoagulation therapy. Considering this, anticoagulants were included in the initial targets of the ADE action plan along with antidiabetics and opioids [4] which eventually led to the foundation of the anticoagulation stewardship program (ASP). In 2019, the US Anticoagulation Forum published a guide on the essential elements of the ASP with the aim to promote effective use of anticoagulants across all healthcare institutions [7]. One study each from the US and Canada further laid down the basis of pharmacist-led ASP through evidence of feasibility, acceptance by the prescribing physicians, and better patient outcomes [8,9]. The ASP, which is referred to as an integrated, effective, and system- related measure to optimize anticoagulant use and prevent avoidable ADEs, was successfully implemented in the US with documented cost savings and better patient outcomes [7,10–12]. Furthermore, various studies in the United Kingdom, Australia,

and Canada also documented the positive impact of ASP on maintaining high standards of care and enhancing hospital medication safety protocols [8,13,14].

The implementation of ASP in low- and middle-income countries (LMICs), including Pakistan, encounters distinct challenges attributable to constraints in resources, discrepancies in healthcare infrastructure, and restricted access to specialized medical services [15,16]. According to the 2019 Global Burden of Disease study, the estimated incidence of cardiovascular diseases in Pakistan was around 918 per 100,000 population, and the estimated mortality rate was around 358 per 100,000 population [17]. Various studies reported that one-fifth of middle-aged adults in Pakistan were at risk of developing coronary artery disease, whereas overall cardiovascular diseases were responsible for around 27% of the total deaths [18,19]. Despite the excessive use of anticoagulants and the risk of serious adverse effects, numerous studies from Pakistan revealed that there were no standard guidelines for anticoagulant management, and a holistic approach focusing on continuum of care, quality improvement, and addressing patient-, clinician-, and healthcare system-level barriers was missing [7,18,20]. These gaps contributed to higher rates of anticoagulant-related complications, hospital readmissions, and unnecessary healthcare costs, all placing an additional burden on an already strained healthcare system in Pakistan [21]. Implementation of ASP in Pakistani hospitals could address these challenges by establishing a standardized, evidence-based framework for the safe and effective use of anticoagulants.

We conducted the first Delphi study in Pakistan, which presented extensive consensus points specifically from cardiologists for implementing ASP in Pakistan. The Delphi approach is a credible agreement technique of collecting expert views, which includes a confidential stepwise procedure involving multiple rounds of feedback to achieve consensus among Delphi experts [22].

## Methodology

### Study design

This study was a countrywide, multicenter, three-round Delphi study to obtain expert views on implementing ASP in Pakistan. It was entirely centered on the responses given by the Delphi experts related to significant issues with the implementation of the ASP [22]. The Delphi technique is a broadly recognized scientific process of organized and systematic knowledge gathering from a group of experts (i.e., a Delphi panel) on contentious or complicated issues using a series of questionnaires [23]. All panel experts offer their insights independently and confidentially, thereby mitigating the potential influence of dominant personalities or collective pressure [24,25]. The Delphi process concludes upon reaching a consensus regarding the topics under discussion. In this study, a modified Delphi method was employed, as outlined by the RAND Corporation and the University of California, Los Angeles (UCLA) recommendations [26]. The Delphi process was conducted in a systematic manner comprising seven distinct phases: 1) Comprehensive literature review conducted by the scientific committee; 2) Generation of discussion topics and questionnaire items by the scientific committee during an in-person meeting; 3) Selection of an expert panel for the Delphi process; 4) Invitation extended to potential participants to take part in the Delphi process; 5) Evaluation of the domain and item set by the expert panel through two iterative rounds utilizing an online platform, adhering to a two-round Delphi methodology; 6) Final deliberation among the experts regarding items that did not gain the expert consensus in the previous rounds; 7) Final consensus analysis (Fig 1).

Data were collected between June and December 2023, and the ethical approval was obtained through the Pharmacy Research Ethics Committee (PREC) by the Islamia University of Bahawalpur, Pakistan (Reference: 166-2023-/PHEC, dated Feb 09, 2023).

### Delphi process

**Selection of Delphi questionnaire items.** The scientific committee conducted a systematic review of the literature, encompassing pertinent clinical guidelines, and assessments related to the management of anticoagulation and the

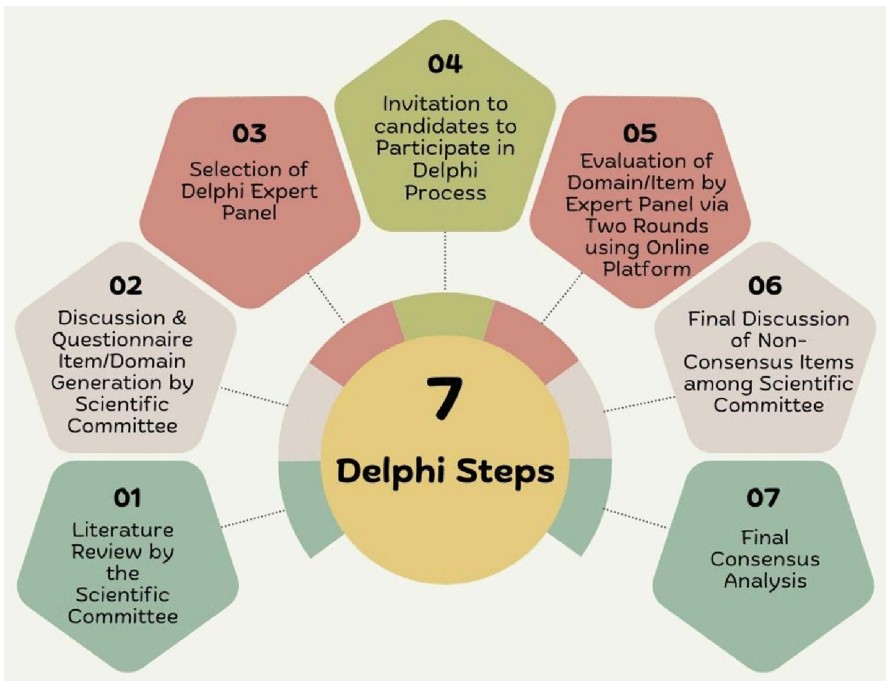

**Fig 1. Delphi steps.**

impact of anticoagulation stewardship on patient health outcomes [8,27–29]. After a thorough and critical examination of the pertinent literature and drawing upon their expertise in anticoagulation, the scientific committee convened in-person meetings to formulate the initial array of domains and items for the Delphi survey questionnaire [25].

**Selection of Delphi participants.** The expert scientific committee comprised five members, including three cardiologists and two professors who were experienced and recognized in anticoagulation therapy management. In total, 90 cardiologists from various hospitals across Pakistan were approached initially and asked to participate in the study as the Delphi panel. The email IDs of target participants were obtained from publicly accessible resources, peers, published academic manuscripts, and/or institutional websites. The selection of experts was based on their substantial expertise and understanding of anticoagulation treatment. During the study, the participants were told that their participation may include numerous rounds (e.g., Round 1 and Round 2), with each round expected to last approximately 15–20 minutes. During Round 1, the participants were also requested to provide their demographic information (i.e., age, gender, type of hospital, hospital position, research experience, and work experience). The participants did not receive any compensation for their time or involvement in the study. Before sending the Delphi survey to all participants, each round was piloted with 5–6 members from the expert group to confirm that the information and queries were comprehensible. The Delphi expert panel also received participant's information pack (PIC) (i.e., participant information sheet, informed consent, and informed consent declaration – S1 File), a detailed leaflet that explained the study's objectives and procedures and an online link to the survey. Online participation in survey meant that they have read PIC and leaflet explaining the study objectives. Ethics committee (PREC) approved this consent procedure.

In the absence of agreement regarding the ideal size of a Delphi group, it is frequently observed that groups consisting of 15–50 participants are sufficient [30].

**Round 1: Idea generation.** The members of the Delphi expert panel were asked to rate their level of agreement with each questionnaire item on a 3-point Likert scale (agree/neutral/disagree) (questionnaire developed in the first step of the

Delphi process). Panelists were additionally prompted to offer remarks on each item. The scientific committee convened face-to-face meetings during which the findings of the survey were presented and deliberated. The selection of items for further deliberations was determined by the endorsement of items by 66.6% of the Delphi expert panel, in conjunction with the consensus of the scientific committee [22].

The items that did not gain a 66.6% level of agreement were either eliminated or revised according to the feedback received from the Delphi expert panel [22]. Following the completion of Round 1 and the subsequent summarization of Delphi expert feedback, modifications were carried out for certain items within the questionnaire. Redundant statements were consolidated and streamlined, while issues related to the clarity of specific statements were addressed based on Delphi expert input. Additionally, new items were created and incorporated as deemed necessary. The revised survey questionnaire was then distributed again to the panelists for Round 2.

**Round 2: Developing consensus.** In Round 2, the same cardiologists (Delphi expert panel) were requested to assess the set of revised/amended items from Round 1 by applying the same voting procedure as in the prior round. For this assessment, the cardiologists received a summary of the anonymously expressed views from their peers (Round 1) and any supplementary data that the scientific committee considered pertinent for facilitating consensus among the panelists. Therefore, the panelists could reflect on the group's answers following the first round and reconsider the non-consensus items based on the views of other panelists. Following the analysis of Round 2 responses, the statements that did not gain experts' consensus were kept for further discussion in Round 3 [23].

**Round 3: Confirming consensus.** This round comprised face-to-face meetings among the scientific committee to evaluate those statements that failed to gain agreement in Round 2 [31]. The participants of the scientific committee deliberated the non-consensus statements till an agreement was achieved to keep or remove these items from the final document.

### Statistical analysis

SPSS 20.0 (IBM Statistics, Armonk, NY, United States) was used for data analysis. Descriptive statistical analysis such as median (range), frequencies and percentage were performed to describe characteristics of the Delphi experts.

## Results

### Delphi panel

Of the 90 cardiologists from 30 hospitals, 75 (83% response rate) agreed to participate in the first round of the Delphi study. Of these, 68 cardiologists participated in the Round 2 assessment. All participants were male with a median age and work experience of 48 and 12 years, respectively. Most of the participants (n = 63, 84%) were from the public sector hospitals (Table 1).

The panel members assessed 33 items from the following 4 domains: 1) Essential Components Needed to Strengthen an Anticoagulant Stewardship Program (11 items), 2) Current Status of Anticoagulant Stewardship Program in Pakistan (5 items), 3) Impact of an Anticoagulant Stewardship Program (9 items), 4) Constraints in the Implementation of the Antico-agulant Stewardship Program (8 items) (See S1 Table).

### Round 1

In Round 1, a total of 18 items (54%) out of 33 items reached a consensus point without any modification. Eight out of eleven items (72.7%) from domain 1, "Essential Components Needed to Strengthen an Anticoagulant Stewardship Program"; one out of five items (20%) from domain 2, "Current Status of Anticoagulant Stewardship Program in Pakistan"; six out of nine items (66.6%) from domain 3, "Impact of an Anticoagulant Stewardship Program"; three out of eight items (60%) from domain 4, "Constraints in the Implementation of the Anticoagulant Stewardship Program" reached a consensus after getting 66.6% acceptance by the Delphi expert panel (n = 75).

**Table 1. Description of the Delphi expert participants.**

| Characteristics | Value |
|---|---|
| **Age (years), median in range** | 48 (35-63) |
| **Gender, male, n (%)** | 75(100) |
| **City, n (%)** | |
| Lahore | 12 (16) |
| Karachi | 15 (20) |
| Islamabad | 9 (12) |
| Rawalpindi | 7 (9.3) |
| Multan | 10 (13.3) |
| Bahawalpur | 10 (13.3) |
| Rahim Yar Khan | 8 (10.7) |
| Sukkur | 4 (5.3) |
| **Professional experience, median (range), years** | 12 (2-25) |
| **Position, n (%)** | |
| Professor | 26 (34.7) |
| Associate Professor | 13 (17.3) |
| Assistant Professor | 27 (36) |
| Other | 9 (12) |
| **Research experience, n (%)** | 55 (73.3) |
| **Type of hospital, n (%)** | |
| Public | 63 (84) |
| Private | 12 (16) |

Note. Participant characteristics were obtained in Round 1 (n = 75). Seven participants did not complete the Round 2 (n = 68).

A total of 15 non-consensus items were identified after Round 1. Among these, four items were thematically similar and were merged into two new items by the expert committee after considering feedback from the Delphi expert panel (cardiologist). The first merged item -merger of item 10 (Domain 1) and item 16 (Domain 2) -resulted in the statement:

"The existing practices on anticoagulants are recommended by the cardiologists, suggesting no need for an ASP program." However, this merged item failed to reach consensus even after modification and was therefore removed. The second merged item -merger of item 28 and item 30 (both from Domain 4) -resulted in the statement:

"The inconsistent guidelines and institutional policies make it difficult to implement ASP." This item was retained for reassessment in Round 2. In total, five non-consensus items that achieved a consensus of less than 40% were removed, including the first merged item (S2 Table). The remaining 8 non-consensus items, comprising seven original and one merged item, were carried forward to Round 2 (S3 Table).

## Round 2

A total of 68 Delphi experts participated in Round 2. During the process, 8 (7 original and 1 merged) non-consensus items in Round 1 were again presented to the cardiologists for further discussion and agreement. Out of these, 4 items finally reached an agreement in Round 2 after a detailed discussion with the scientific committee. Four consensus items included 2 items from domain 1, "Essential components needed to strengthen an ASP" and one item each from domain 3, "Impact of an ASP" and domain 4 "Constraints in the implementation of the ASP". The remaining four non-consensus items (items 25, 32, 27, and 29) were taken to the next round for further discussion.

### Round 3

Four items that failed to reach consensus in Round 2 were deliberated in Round 3 by the scientific committee, where two items were finally accepted (i.e., one item each from domain 3 "Impact of an ASP" and domain 4, "Constraints in the implementation of the ASP"). Two items were rejected from Domain 4 (items 27 and 29). After the Delphi process was finalized, 24 consensus points were ultimately selected. Fig 2 shows the results of the Delphi study.

The final 24 items included 10 statements for domain 1, one statement for domain 2, eight statements for domain 3, and five statements for domain 4. Table 2 summarizes the findings from cardiologists.

## Discussion

To the best of our knowledge, this is probably the first study from Pakistan that described the potential for implementation of ASP in Pakistan. Globally, the implementation of ASP improved clinical outcomes, promoted patient safety, and supported healthcare systems in delivering high-quality, evidence-based anticoagulation management [7,8,11–13,32]. Using a Delphi technique, we developed consensus-driven recommendations for the implementation of ASP with an aim to promote rational use of anticoagulation therapy in Pakistani healthcare settings.

The present study indicated absolute agreement among Pakistani cardiologists regarding the implementation of ASP in Pakistan. A considerable consensus was achieved about the essential components needed to strengthen ASP for anticoagulation management in the cardiology departments. The expert views were generally in line with the positive impact of ASP on patients' morbidity and mortality rates associated with cardiovascular diseases. However, this study revealed that there were some constraints regarding the implementation of ASP, which must be optimized based on the standard recommendations [7,12].

More than 90% of cardiologists agreed that hospital staff should collaborate with health authorities and there must be representatives from key areas such as pharmacy, data analysis, and administrative leadership for the successful implementation of ASP. This statement is strongly endorsed in the literature [2,33,34]. A recent study from the US further elaborated that the essential step to promote, implement, and sustain ASP was to secure partnership with administrative leadership [35]. The existing literature also evidenced educational sessions with cardiologists, comprehensive education among patients, and training of ASP team for successful implementation of ASP in healthcare settings [8,27,28]. This is in line with the findings of our study. The agreement of our study participants regarding the availability of a checklist for standardized core elements of ASP in hospitals further highlighted the need to foster optimal anticoagulation management, which is consistent with the published literature [7,12]. Pakistan is a resource-limited country that only spends 2.9% of its GDP on healthcare [36]. Therefore, investment in value-added programs such as ASP is minimal. However, there is a need to invest in healthcare programs, which may result in future cost savings [28]. The same is being reported by our study participants.

There was considerable agreement among study participants on the positive impact of ASP on the patients and the healthcare system. They endorsed that this program not only focuses on patient health outcomes, such as reducing ADRs (severe bleeding events and thromboembolic events), promoting routine INR monitoring, and decreasing morbidity and mortality rates, but also decreases the workload on healthcare providers, as consistently noted in literature [28]. Moreover, our study participants also agreed that ASP provide key training and information opportunities to healthcare providers and patients, thereby improving treatment decisions and patient adherence. This is consistent with recently published studies on the need for essential ASP educational trainings [33,37,38]. Continuous medical education is a part of routine clinical practice in Pakistan; however, training opportunities specific to the needs of ASP are lacking.

The Delphi panel unanimously agreed that the regulatory authorities and policymakers were not focused on the implementation of ASP, which is also reported in the recent studies [8,22,39]. The panelists further stated that the inconsistent guidelines and institutional policies made it difficult to implement ASP. Evidence-based implementation of ASP in Pakistan

**Round 1**
**(June 2 - August 3, 2023)**
**75 Cardiologists**

- 33 items divided across 4 domains.
- A total of 18 items (54%) achieved consensus and required no modifications.
  - ✓ Essential Components Needed to Strengthen an Anticoagulant Stewardship Program: 8/11 items.
  - ✓ Current Status of Anticoagulant Stewardship Program in Pakistan: 1/5 items.
  - ✓ Impact of an Anticoagulant Stewardship Program: 6/9 items.
  - ✓ Constraints in the Implementation of the Anticoagulant Stewardship Program: 3/8 items.
- 15 non-consensus items.
  - ✓ 2 items merged from 4 original items.
  - ✓ 5 items eliminated, including 1 merged item.
- 8 non-consensus items included in Round 2 (7 original + 1 merged item).

**Round 2**
**(August 25 -October 27, 2023)**
**68 Cardiologists**

- 8 items which failed to reach consensus or were modified in Round 1.
- Consensus was achieved on 4 items.
  - ✓ Essential Components Needed to Strengthen an Anticoagulant Stewardship Program: 2/2 items.
  - ✓ Current Status of Anticoagulant Stewardship Program in Pakistan: 0/0 items.
  - ✓ Impact of an Anticoagulant Stewardship Program:1/2 items.
  - ✓ Constraints in the Implementation of the Anticoagulant Stewardship Program: 1/4 items.
- 4 items to be evaluated in Round 3.

**Round 3**
**(November 1 - December 15, 2023)**
**Scientific Committee**

- Review of 4 items that failed to reach consensus in Round 2.
- 2 items reached consensus.
  - ✓ Impact of an Anticoagulant Stewardship Program:1/1 item.
  - ✓ Constraints in the Implementation of the Anticoagulant Stewardship Program: 1/3 items.
- 2 items rejected from Domain 4.

**Final document**
24 consensus items

**Fig 2. Results of the Delphi study.**

**Table 2. Final round results (Endorsed best points in ranked order from most to least important within each domain. See footnote for terminology and definitions.)[a].**

**Domain 1. Essential components needed to strengthen an Anticoagulant Stewardship Program.**

| Statement no. | Domain 1. Statements | Consensus (%) | Missing Count[b] | Consensus Reached at Round |
|---|---|---|---|---|
| 03 | The hospitals should collaborate with health authorities regarding the implementation of the Anticoagulant Stewardship Program. | 97 | 0 | 1 |
| 02 | In hospitals, local anticoagulant guidelines should be developed to promote the Anticoagulant Stewardship Program. | 86 | 0 | 1 |
| 09 | There must be representatives from key areas to obtain valuable viewpoints from all domains of the care delivery system such as pharmacy, data analysis, and administrative leadership for the successful implementation of the Anticoagulant Stewardship Program. | 88 | 1 | 1 |
| 01 | The leader or co-leaders, such as physicians and pharmacists, must be appointed for Anticoagulant Stewardship Program management and its outcomes. | 85 | 0 | 1 |
| 04 | Educational sessions with cardiologists in the hospitals will help to raise awareness of the Anticoagulant Stewardship Program and encourage discussion on the importance of actions regarding the appropriate anticoagulant use. | 90 | 0 | 1 |
| 05 | Financial resources should be allocated for the development of anticoagulant stewardship activities. | 85 | 0 | 1 |
| 06 | Comprehensive education for patients, their families, and caregivers should be essential for ensuring safe and effective management of anticoagulant therapy. | 88 | 0 | 1 |
| 08 | The clinical experts are responsible for anticoagulant stewardship and should achieve their expertise level through advanced training. | 85 | 0 | 1 |
| 07 | The checklist for standardized core elements of the Anticoagulant Stewardship Program should be available in hospitals. | 84 | 4 | 2 |
| 11 | Anticoagulant guidelines must be electronically accessible to all. | 80 | 0 | 2 |

**Domain 2. Current Status of Anticoagulant Stewardship Program in Pakistan**

| Statement no. | Domain 2. Statements | Consensus % | Missing Count[b] | Consensus reached at round |
|---|---|---|---|---|
| 15 | The implementation of the Anticoagulant Stewardship Program (ASP) should be recommended in hospitals or cardiac wards. | 100 | 0 | 1 |

**Domain 3. Impact of an Anticoagulant Stewardship Program.**

| Statement no. | Domain 3. Statements | Consensus % | Missing Count[b] | Consensus reached at round |
|---|---|---|---|---|
| 18 | The successful implementation of the Anticoagulant Stewardship Program will improve patients' health outcomes. | 97 | 0 | 1 |
| 21 | The Anticoagulant Stewardship Program will reduce adverse drug reactions such as severe bleeding events and thromboembolic events. | 91 | 0 | 1 |
| 17 | The Anticoagulant Stewardship Program implementation will have a positive impact on the healthcare system in Pakistan. | 85 | 0 | 1 |
| 22 | The Anticoagulant Stewardship Program (ASP) will reduce healthcare expenditures in the healthcare system. | 82 | 0 | 1 |
| 23 | The Anticoagulant Stewardship Program will provide key training and information to healthcare providers and patients, improving treatment decisions and patient adherence. | 91 | 0 | 1 |
| 20 | The Anticoagulant Stewardship Program will contribute to the reduction of morbidity and mortality rates associated with cardiovascular diseases. | 85 | 0 | 1 |
| 19 | The Anticoagulant Stewardship Program will promote routine INR monitoring according to patients' indications. | 82 | 0 | 2 |
| 25 | The implementation of the Anticoagulant Stewardship Program may decrease the workload and burden for healthcare providers. | 80 | 3 | 3 |

*(Continued)*

**Table 2.** (Continued)

| Statement no. | Domain 1. Statements | Consensus (%) | Missing Count[b] | Consensus Reached at Round |
|---|---|---|---|---|
| **Domain 1. Essential components needed to strengthen an Anticoagulant Stewardship Program.** | | | | |

| Statement no. | Domain 4. Statements | Consensus % | Missing Count[b] | Consensus reached at round |
|---|---|---|---|---|
| **Domain 4. Constraints in the implementation of the Anticoagulant Stewardship Program.** | | | | |
| 26 | The regulatory authorities and policymakers are not focused on the implementation of an Anticoagulant Stewardship Program. | 94 | 0 | 1 |
| 31 | Current training for healthcare professionals on anticoagulant management is insufficient and requires enhancement. | 88 | 0 | 1 |
| 33 | Financial limitations have a significant effect on the successful implementation of the Anticoagulant Stewardship Program. | 85 | 0 | 1 |
| 28 | The inconsistent guidelines and institutional policies make it difficult to implement Anticoagulant Stewardship Programs. | 82 | 2 | 2 |
| 32 | Cardiologists find it difficult to hold annual meetings on anticoagulant management, as such meetings are not typically conducted, making this system difficult to implement. | 80 | 0 | 3 |

[a]**Terminology and definitions used throughout the statements 'Anticoagulant Stewardship Program (ASP**)' are defined as a coordinated, efficient, and sustainable system-level initiative designed to achieve optimal anticoagulant-related health outcomes and minimize avoidable ADEs. It is successful in improving health outcomes and minimizing avoidable ADEs. **'INR'** stands for "International Normalized Ratio".

[b]Missing indicates that cardiologists indicated insufficient knowledge to scale that item.

is problematic due to limited industry-academia linkage. Moreover, healthcare programs and departments seldom conduct operational and implementation research (OR and IR) for process improvement. This is partly due to a lack of trained OR staff and limited funding opportunities for trainings, workshops, and conferences. The participants in this study also agreed that lack of funding led to limited training and educational activities.

### Impact of findings on policy and practice

This Delphi study presents consensus points specifically from cardiologists for the implementation of ASP in Pakistan. To overcome the barriers associated with the implementation of ASP, several key components were proposed. The most effective way to address the situation was to allocate financial resources for the development of anticoagulation stewardship activities. Additionally, since the clinicians are one of the key people in the management of anticoagulants, therefore, they must have training opportunities to improve their knowledge and skills. It is also recommended that electronic guidelines for the healthcare workforce should be developed that are easily accessible and provide up-to-date protocols for anticoagulation management. Additionally, a healthcare facility-specific core elements checklist should be created to improve patient outcomes, ensuring that all parameters are consistently monitored during the treatment. This approach would streamline the decision-making and improve the quality of care.

From a practice perspective, the implementation of this program is necessary to optimize the anticoagulation therapy, minimize the escalating adverse drug reaction-related issues, reduce morbidity and mortality rates, and decrease the burden on healthcare providers. In order to achieve these objectives, a multidisciplinary team effort is required. It will be crucial to gather the perspectives of all relevant stakeholders, such as clinicians, pharmacists, patients, and policymakers, to comprehend the possible challenges of implementing this program in Pakistan. Further, studies can also focus on the implementation of ASP through the IR cycle and see how it works.

## Strengths and limitations

This study has notable strengths, including a high response rate, keen involvement of cardiologists, and a high agreement rate. The comparatively cohesive opinions manifested by the cardiologists suggest that the findings include extensive relevance to the implementation of ASP. Further strengths comprise the execution of protocols aligned with standard Delphi frameworks, including confidentiality of feedback, open-ended statement creation and progressive development of items, and stability assessments of items in various rounds.

There are a few limitations to the study. First, the findings of the study may not be generalized for other countries based on the fact every country has her own healthcare system and needs. Second, only cardiologists were part of this study, while the viewpoints of policymakers and other healthcare professionals such as pharmacists, nurses, etc., were not obtained. We did not include other healthcare professionals because their inclusion was not meaningful due to their limited role in clinical and administrative matters of healthcare delivery. Third, this study did not evaluate the feasibility and affordability of implementing ASP under this project.

## Conclusion

This Delphi study developed 24 consensus points for implementing ASP in Pakistan. The Delphi members suggested the essential components needed to strengthen ASP. This study acknowledged strategies to implement ASP, including educational sessions for patients and healthcare professionals, collaboration with healthcare authorities, and allocation of financial resources. This study also identified consensus of cardiologists on the perceived benefits of ASP for patients and the healthcare system. Several barriers that have hindered the implementation of ASP in Pakistan were identified, including patient- and healthcare system-related barriers.

## Supporting information

**S1 File. Participant information pack.**
(DOCX)

**S1 Table. Data collection tool for cardiologists.**
(DOCX)

**S2 Table. Items eliminated in the first round of the Delphi process.**
(DOCX)

**S3 Table. Non-consensus items carried forward to Round 2 and onwards.**
(DOCX)

## Acknowledgments

The authors are thankful to the cardiologists for participating in the study.

## Author contributions

**Conceptualization:** Wajiha Razzaq, Muhammad Atif.

**Data curation:** Wajiha Razzaq, Kanza Arshad.

**Formal analysis:** Wajiha Razzaq, Muhammad Atif, Kanza Arshad, Imran Masood.

**Investigation:** Wajiha Razzaq.

**Methodology:** Wajiha Razzaq, Muhammad Atif, Kanza Arshad, Imran Masood.

**Project administration:** Wajiha Razzaq, Kanza Arshad.

**Resources:** Wajiha Razzaq.

**Supervision:** Muhammad Atif, Imran Masood.

**Writing – original draft:** Wajiha Razzaq, Kanza Arshad, Imran Masood.

**Writing – review & editing:** Muhammad Atif.

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
