## [Decision Letter · Decision Letter 0]

30 Apr 2025

Dear Dr. Atif,

Thank you for submitting your manuscript to PLOS ONE. After careful consideration, we feel that it has merit but does not fully meet PLOS ONE’s publication criteria as it currently stands. Therefore, we invite you to submit a revised version of the manuscript that addresses the points raised during the review process.

We look forward to receiving your revised manuscript.

Kind regards,

Moises Auron, MD, FAAP, FACP, SFHM, FRCP (Lon), FRCPCH

Academic Editor

PLOS ONE

Journal Requirements:

2. In the online submission form, you indicated that “Data shall be made available by the corresponding author upon receiving reasonable request through editor in chief.”

3.Please review your reference list to ensure that it is complete and correct. If you have cited papers that have been retracted, please include the rationale for doing so in the manuscript text, or remove these references and replace them with relevant current references. Any changes to the reference list should be mentioned in the rebuttal letter that accompanies your revised manuscript. If you need to cite a retracted article, indicate the article’s retracted status in the References list and also include a citation and full reference for the retraction notice.

Reviewers' comments:

Reviewer's Responses to Questions

**Comments to the Author**

1. Is the manuscript technically sound, and do the data support the conclusions?

Reviewer #1: Yes

2. Has the statistical analysis been performed appropriately and rigorously?

Reviewer #1: Yes

3. Have the authors made all data underlying the findings in their manuscript fully available?

Reviewer #1: No

4. Is the manuscript presented in an intelligible fashion and written in standard English?

Reviewer #1: Yes

Reviewer #1: Kudos to this article, it represents great work.

Some suggestions and things to consider:

I am a little confused about the focus of the paper, is there more of a focus on the modified Delphi technique or the ASP?

Since I think the key element of the paper is the use of the modified Delphi technique, I think the introduction needs some background on the modified Delphi technique and whether it has been used in other similar programs. There could also be more mentions of similar use in the discussion part of the paper.

Page 7: restructure the line "In total, there 90 cardiologist...."

Page 18: Please restructure the line: "Although this program focuses on patient health outcomes, such as reducing

adverse drug reactions (severe bleeding events and thromboembolic events), promoting routine

INR monitoring according to patients’ indications, reduction of morbidity and mortality rates,

being consistently identified in the literature, as well as decrease the workload and burden for

healthcare providers"

Page 21: In the conclusion, I think it is inaccurate to state that the study identified the impact of an ASP on benefits for patients and the healthcare system; you could state that it identified a cardiology consensus on the perceived benefits for patients etc.

Can you also list the 5 elements/items that were eliminated in the first round of the Delphi process

Finally, I wonder if the entire item list for the ASP Delphi needs to be considered a cardiologists' perception of benefits and constraints, impact, ideal implementation in the discussion since no pharmacists, other policymakers were involved in the making of the ASP and the implementation of on-the-ground policies such as these need multi-specialty collaboration and expertise.

**Do you want your identity to be public for this peer review?** For information about this choice, including consent withdrawal, please see our Privacy Policy

Reviewer #1: **Yes: ** Arunab Mehta, MD, MEd

---

## [Author Response · Author response to Decision Letter 1]

26 Jun 2025

Response to reviewer comments is provided as an attachment.

Response to Editorial Comments.

1. In the online submission form, you indicated that “Data shall be made available by the corresponding author upon receiving reasonable request through editor in chief.”

Author response: Amended as required.

2. Please provide additional details regarding participant consent. In the Methods section, please ensure that you have specified (1) whether consent was informed and (2) what type you obtained (for instance, written or verbal). If your study included minors, state whether you obtained consent from parents or guardians. If the need for consent was waived by the ethics committee, please include this information.

Author response: Manuscript amended as per need. Supplementary File 1 provided in support.

---

## [Editor Report · Decision Letter 1]

3 Jul 2025

Dear Dr. Atif,

Thank you for submitting your manuscript to PLOS ONE. After careful consideration, we feel that it has merit but does not fully meet PLOS ONE’s publication criteria as it currently stands. Therefore, we invite you to submit a revised version of the manuscript that addresses the points raised during the review process.

We look forward to receiving your revised manuscript.

Kind regards,

Moises Auron, MD, FAAP, FACP, SFHM, FRCP (Lon), FRCPCH

Academic Editor

PLOS ONE

Journal Requirements:

Additional Editor Comments:

Thank you for your submission and updates. We recommend a minor revision with the following main recommendations:

Kudos to this article, it represents great work.

Some suggestions and things to consider:

I am a little confused about the focus of the paper, is there more of a focus on the modified Delphi technique or the ASP?

Since I think the key element of the paper is the use of the modified Delphi technique, I think the introduction needs some background on the modified Delphi technique and whether it has been used in other similar programs. There could also be more mentions of similar use in the discussion part of the paper.

Page 7: restructure the line "In total, there 90 cardiologist...."

Page 18: Please restructure the line: "Although this program focuses on patient health outcomes, such as reducing

adverse drug reactions (severe bleeding events and thromboembolic events), promoting routine

INR monitoring according to patients’ indications, reduction of morbidity and mortality rates,

being consistently identified in the literature, as well as decrease the workload and burden for

healthcare providers"

Page 21: In the conclusion, I think it is inaccurate to state that the study identified the impact of an ASP on benefits for patients and the healthcare system; you could state that it identified a cardiology consensus on the perceived benefits for patients etc.

Can you also list the 5 elements/items that were eliminated in the first round of the Delphi process

Finally, I wonder if the entire item list for the ASP Delphi needs to be considered a cardiologists' perception of benefits and constraints, impact, ideal implementation in the discussion since no pharmacists, other policymakers were involved in the making of the ASP and the implementation of on-the-ground policies such as these need multi-specialty collaboration and expertise.

We look forward to receive your updated version.

---

## [Author Response · Author response to Decision Letter 2]

18 Jul 2025

1. Response to reviewers is provided as an attachment.

2. The Editor has asked use to respond to same comments which we have addressed in first revision. Please advice how to proceed.

---

## [Decision Letter · Decision Letter 2]

12 Nov 2025

Establishing consensus on the implementation of Anticoagulation Stewardship Program with cardiologists in Pakistan: A Delphi study

PONE-D-25-01990R2

Dear Dr. Atif,

We’re pleased to inform you that your manuscript has been judged scientifically suitable for publication and will be formally accepted for publication once it meets all outstanding technical requirements.

Kind regards,

Moises Auron, MD, FAAP, FACP, SFHM, FRCP (Lon), FRCPCH

Academic Editor

PLOS ONE

Additional Editor Comments (optional):

Reviewers' comments:

Reviewer's Responses to Questions

**Comments to the Author**

Reviewer #2: (No Response)

2. Is the manuscript technically sound, and do the data support the conclusions?

Reviewer #2: Yes

3. Has the statistical analysis been performed appropriately and rigorously?

Reviewer #2: Yes

4. Have the authors made all data underlying the findings in their manuscript fully available?

Reviewer #2: Yes

5. Is the manuscript presented in an intelligible fashion and written in standard English?

Reviewer #2: Yes

Reviewer #2: study is coherently synthesized and described appropriately.

findings are shared in articulate fashion and important clinical area is highlighted.

**Do you want your identity to be public for this peer review?** For information about this choice, including consent withdrawal, please see our Privacy Policy

Reviewer #2: **Yes: ** Mohammad Mohmand

---

## [Editor Report · Acceptance letter]

PONE-D-25-01990R2

PLOS ONE

Dear Dr. Atif,

I'm pleased to inform you that your manuscript has been deemed suitable for publication in PLOS ONE. Congratulations! Your manuscript is now being handed over to our production team.

Kind regards,

on behalf of

Dr. Moises Auron

Academic Editor

PLOS ONE